# EMB-GAM: AN INTERPRETABLE AND EFFICIENT PREDICTOR USING PRE-TRAINED LANGUAGE MODELS

## ABSTRACT

Deep learning models have achieved impressive prediction performance but often sacrifice interpretability and speed, critical considerations in high-stakes domains and compute-limited settings. In contrast, generalized additive models (GAMs) can maintain interpretability and speed but often suffer from poor prediction performance due to their inability to effectively capture feature interactions. This work aims to bridge this gap by using pre-trained neural language models to extract embeddings from each input before aggregating them and learning a linear model in the embedding space. The final model (which we call Emb-GAM) is a transparent, linear function of its input features and feature interactions. Leveraging the language model allows Emb-GAM to learn far fewer linear coefficients, model larger interactions, dramatically speed up inference, and generalize well to novel inputs (e.g. unseen ngrams in text). Across a variety of natural-language-processing datasets, Emb-GAM achieves strong prediction performance without sacrificing interpretability or speed. All code is made available on Github.

## 1 INTRODUCTION

Large neural language models (LLMs) have demonstrated impressive predictive performance due to their ability to learn complex, non-linear, relationships between variables. However, the inability of humans to understand these relationships has led LLMs to be characterized as black boxes, often limiting their use in high-stakes applications such as science (Angermueller et al., 2016), medicine (Kornblith et al., 2022), and policy-making (Brennan & Oliver, 2013). Moreover, the use of black-box models such as LLMs has come under increasing scrutiny in settings where users require explanations or where models struggle with issues such as fairness (Dwork et al., 2012) and regulatory pressure (Goodman & Flaxman, 2016). Simultaneously, recent black-box models have grown to massive sizes, making them costly and difficult to deploy, particularly for edge devices such as mobile phones.

As an alternative to large black-box models, transparent models, such as generalized additive models (Hastie & Tibshirani, 1986) and rule-based models (Breiman et al., 1984) can maintain interpretability. Additionally, transparent models tend to be faster and more computationally efficient than black-box models. While transparent models can sometimes perform as well as black-box models (e.g. Rudin et al. (2021); Ha et al. (2021); Mignan & Broccardo (2019); Tan et al. (2022)), in many settings such as natural-language processing (NLP), there is often a large gap in the performance between transparent models and black-box models.

This work aims to minimize this gap by leveraging a pre-trained LLM to learn a more effective transparent model. Specifically, we extract LLM embeddings for different feature interactions (e.g. ngrams in text) and then learn a generalized additive model on top of these embeddings. The final learned model (which we call Emb-GAM) is a transparent, linear function of its input features and feature interactions, but the use of the LLM allows Emb-GAM to intelligently reduce its number of learned parameters (see Fig 1). Rather than learning a linear model over all possible feature interactions (which scales exponentially with the order of the interaction and the feature dimension), Emb-GAM requires learning only a fixed set of linear coefficients (the size of the embedding extracted by the LLM).

As a result, Emb-GAM can efficiently model high-order interactions, generalize well to novel interactions, and even vary the number of features used at test-time for prediction. Moreover, inference

with Emb-GAM is extremely fast, requiring only looking up coefficients from a dictionary and then summing them. Experiments on a variety of NLP classification datasets show that Emb-GAM can achieve better generalization accuracy than transparent baseline methods. Moreover, learned Emb-GAM models are easily interpretable, both for individual predictions and at the level of an entire dataset, enabling use cases in high-stakes settings. In what follows, Sec 2 covers the background and related work, Sec 3 explains the Emb-GAM pipeline, Sec 4 shows the results, and Sec 5 concludes with a discussion.

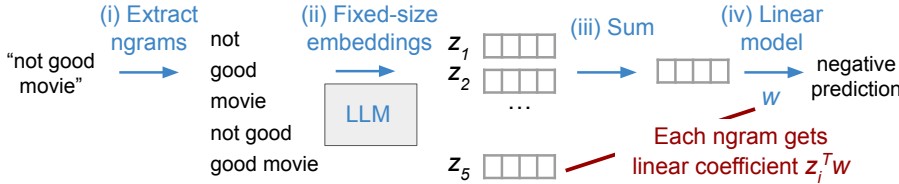

Figure 1: Overview of Emb-GAM. A pretrained neural network is used to extract fixed-size embeddings for ngrams in a given sequence. Then, these embeddings are summed and used to train a supervised linear model. At test time, the model can be interpreted exactly as a generalized additive model, with a linear coefficient for each ngram in the input.

## 2 BACKGROUND AND RELATED WORK

**GAMs**    There is a large literature on additive models being used for interpretable modeling. This includes generalized additive models (GAMs) (Hastie & Tibshirani, 1986) that have evolved to achieve strong performance by modeling individual component functions/interactions using regularized boosted decision trees (Caruana et al., 2015) and more recently using neural networks (Agarwal et al., 2021). While GAMs have been successful in a variety of domains, existing GAM methodology is limited in its ability to model the high-order feature interactions that arise in NLP.

Meanwhile, NLP has seen great success in models which build strong word-level representations, e.g. word2vec (Mikolov et al., 2013a;b), GloVe (Pennington et al., 2014), and ELMo (Peters et al., 2018). By replacing such models with LLM embeddings, Emb-GAM enables easily modeling ngrams of different lengths without training a new model. Moreover, LLMs can incorporate information about labels into the embeddings (e.g. by first finetuning an LLM on a downstream prediction task).

**Other transparent models**    Rule-based methods, such as trees (Breiman et al., 1984; Lin et al., 2020), regularized trees (Agarwal et al., 2022), rule sets (Friedman & Popescu, 2008), lists (Angelino et al., 2017; Singh et al., 2021), and tree sums (Tan et al., 2022) perform well for a variety of tasks, but are often less effective in NLP tasks, where the number of required rules tends to grow with the size of the vocabulary.

An alternative line of work aims to make neural networks more interpretable. For example, models can make predictions by comparing inputs to prototypes (Li et al., 2018; Chen et al., 2019) or by predicting intermediate interpretable concepts (Koh et al., 2020). Alternatively, the entire model can be distilled into a transparent model (e.g. adaptive wavelets (Ha et al., 2021)). However, these works have been largely been restricted to computer vision.

**Feature and feature-interaction importances**    Loosely related to this work are post hoc methods that aim to help understand a black-box model, i.e. by providing feature importances using methods such as LIME (Ribeiro et al., 2016), SHAP (Lundberg et al., 2019), and others (Friedman, 2001; Devlin et al., 2019). However, these post-hoc methods lose some information by summarizing the model and suffer from issues with summarizing interactions (Rudin, 2018; Murdoch et al., 2019). Slightly more related are works which aim to explain feature interactions or transformations in

neural networks (Janizek et al., 2021; Singh et al., 2019; 2020), but these works fail to explain the model as a whole and are again less reliable than having a fully transparent model.

## 3  METHODS: EMB-GAM PIPELINE DESCRIPTION

### 3.1  PRELIMINARIES: GAMS

Generalized additive models, or GAMs (Hastie & Tibshirani, 1986) take the form:

$$g(\mathbb{E}[y]) = \beta + f_1(x_1) + f_2(x_2) + \cdots + f_K(x_K), \tag{1}$$

where $\mathbf{x} = (x_1, x_2, \ldots, x_K)$ is the input with $K$ features. $y$ is the target variable, $g(.)$ is the link function (e.g., logistic function) and each $f_i$ is a univariate shape function with $\mathbb{E}[f_i] = 0$. Due to the function's additivity, each component function $f_i$ can be interpreted independently. Generalized linear models, such as logistic regression, are a special form of GAMs where each $f_i$ is restricted to be linear.

**GAMs fail to effectively capture interactions in NLP**  In natural-language processing, GAMs usually take the form of a *bag-of-words model*, in which each feature $x_i$ is a binary indicator (or count) of the presence of a single token (e.g. the word *good*). However, this model fails to capture interactions between features (e.g. *not good* is different than the sum of *not* and *good*), which is crucial in NLP tasks.

The most common way to deal with interactions is including ngrams as features (a *bag-of-ngrams model*); in this case, each feature is formed by concatenating $n$ tokens, e.g. *not good* would be a 2-gram, also known as a bigram. However, the number of ngrams in a dataset grows exponentially with $n$ and the vocab-size. Even for a modest vocab-size of 10,000 tokens, the number of possible trigrams is already $10^{12}$. This makes it exceedingly difficult to learn accurate coefficients for linear models on ngrams in the training set and impossible to learn coefficients for ngrams not seen in the training set.

### 3.2  THE EMB-GAM PIPELINE

To remedy the issues with the bag-of-ngrams model, we propose Emb-GAM, an intuitive model which leverages a pre-trained language model $\phi$ to extract a better feature representation $z_i = \phi(x_i)$ for each input ngram $x_i$ (see Fig 1). This allows learning only a single linear weight vector $w$ that has the same dimension as $z_i$, regardless of the number of ngrams. The learned model is still a GAM, ensuring that the model can be cleanly interpreted as a linear function of its inputs:

$$g(\mathbb{E}[y]) = \beta + w^T \sum_i \phi(x_i) \tag{2}$$

Emb-GAM consists of four steps, each of which can be modified slightly:

**(i) Extracting ngrams.**  To begin, a user specifies a procedure to extract a set of ngram features from an input text sequence. It is important that the extracted ngrams be semantically meaningful. For NLP, a word-level tokenizer can extract meaningful unigrams from the text; all experiments here use the spaCy tokenizer (Honnibal & Montani, 2017). The order of interaction to be used (i.e. the length of an ngram) can be pre-specified or selected via cross-validation. Note that the longer the included interactions become, the less interpretable the resulting model will be. Domain knowledge can be used to improve the ngram extraction process: for example, common stopwords can be removed as features or language-specific parse trees can be used to extract out key ngrams.

**(ii) Extracting embeddings.**  In the embedding step, each input (i.e. ngram) is fed through the model to retrieve a fixed-size embedding. If a transformer returns a variable-length embedding (e.g. the embedding is the size of the sequence length), we average over its variable-length dimension[1].

---

[1]A common alternative for bi-directional (masked) language models is to use the embedding for a special token (i.e. `[CLS]`), but we aim to keep the approach here more general.

For Emb-GAM to work well, it is important that the pre-training task (e.g. next-word prediction) contains useful information about the interactions which are used in a downstream task (e.g. sentiment classification).

**(iii) Summing embeddings.** In the summation step, the embeddings of each ngram in the input are summed to yield a single fixed-size vector, ensuring additivity of the final model. While we conduct a naive sum here, the sum could be adjusted with weights (e.g. weighting ngrams of different orders differently).

**(iv) Fitting the final linear model to make predictions.** Finally, we obtain a prediction by training a linear model on the summed embedding vector. Importantly, the fitted model in Eq. (2) can still be exactly decomposed into an additive combination of terms: for each ngram $x_i$, its linear contribution to the final prediction is simply $w^T\phi(x_i)$. In the classification experiments performed here, we choose the link function $g$ to be the logit function (or the softmax for multi-class) and also add $\ell_2$ regularization over the parameters $w$ in Eq. (2).

**Computational considerations** Emb-GAM is inexpensive to fit as (i) the pre-trained language model is used only for inference and (ii) it only requires fitting a linear model to relatively few features. After training, the model can be converted to a dictionary of linear coefficients for each ngram, making inference extremely fast. Making a prediction requires simply looking up the scalar coefficient for each ngram in a sample, where the coefficient is the dot product between the ngram's embedding and the learned linear weight $w$, which is cached during training. Nevertheless, it may still be useful to retain the embedding model at test-time to infer coefficients for previously unseen ngrams.

## 4 RESULTS: EMB-GAM PREDICTS WELL WHILE MAINTAINING INTERPRETABILITY

### 4.1 DATA OVERVIEW

In this work, we study four widely used NLP classification datasets spanning different domains including classifying the emotion of tweets (Saravia et al., 2018), the sentiment of financial news sentences (Malo et al., 2014), or the sentiment of movie reviews (Pang & Lee, 2005; Socher et al., 2013) (see Table 1 for an overview).

Table 1: Overview of datasets used here. The number of ngrams explodes with the size of the ngram.

|  | Financial phrasebank | Rotten tomatoes | SST2 | Emotion |
| --- | --- | --- | --- | --- |
| Samples (train) | 2,313 | 8,530 | 67,349 | 16,000 |
| Samples (val) | 1,140 | 1,066 | 872 | 2,000 |
| Classes | 3 | 2 | 2 | 6 |
| Majority class fraction | 0.62 | 0.5 | 0.56 | 0.34 |
| Unigrams | 7,169 | 16,631 | 13,887 | 15,165 |
| Bigrams | 28,481 | 93,921 | 72,501 | 106,201 |
| Trigrams | 39,597 | 147,426 | 108,800 | 201,404 |
| Fraction of trigrams appearing only once | 0.91 | 0.93 | 0.13 | 0.89 |

Across datasets, the number of bigrams and trigams quickly explodes, making it difficult to fit an accurate bag-of-ngrams model. Moreover, many ngrams appear very rarely; for example, in the rotten tomatoes datasets, 93% of trigrams appear only once in the training dataset.

## 4.2 PREDICTION RESULTS

For each dataset, we fit Emb-GAM along with two baselines: a bag-of-ngrams model and the popular TF-IDF model (Jones, 1972).[2] In each case, a model is fit via cross-validation on the training set (to tune the amount of $\ell_2$ regularization added) and its accuracy is evaluated on the validation set.

**Generalization as a function of ngram size**  Fig 2 shows the generalization accuracy of the different models across different datasets as a function of the included ngram size. In this plot, the language model used to extract embeddings $\phi$ is a BERT model (Devlin et al., 2018), finetuned on each individual dataset.[3] Emb-GAM performs well compared to the baselines, achieving a considerable increase in accuracy across three of the four datasets. Notably, Emb-GAM performance tends to increase as higher-order ngrams are added, whereas the baseline methods do not.

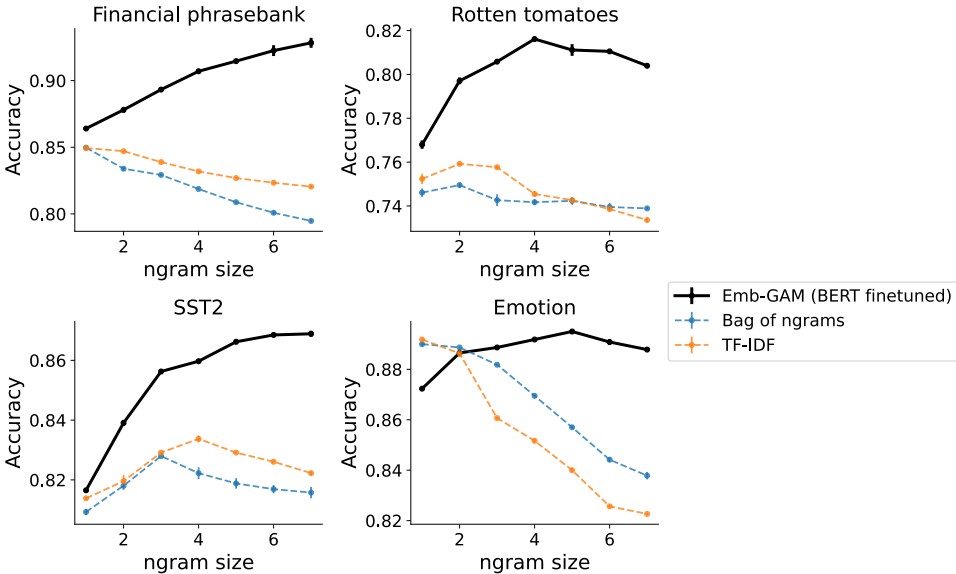

Figure 2: Generalization accuracy as a function of ngram size. As the size of the included ngram increases, the gap between the BERT finetuned model and the original models grows. Averaged over three random cross-validation splits; error bars are standard errors of the mean (many are within the points).

**Best generalization results**  Table 2 shows the best results when choosing the order of ngrams via cross-validation (again using a finetuned BERT model to extract embeddings for Emb-GAM). When comparing to the interpretable baselines (GloVE (Pennington et al., 2014)[4] in addition to Bag of ngrams and TF-IDF), Emb-GAM shows considerable gains on three of the datasets (*Financial phrasebank*, *Rotten tomatoes*, and *SST2*) and only a minor gain on the tweet dataset (*Emotion*), likely because this dataset requires learning less high-order interactions. When restricting Emb-GAM to only use unigrams, performance no longer improves over interpretable baseline methods.

---

[2]The TF-IDF ("Term frequency–inverse document frequency") model is an additive model of ngrams which represents each ngram by its count rescaled by its frequency across the training dataset.

[3]Pre-trained language models are retrieved from HuggingFace (Wolf et al., 2019) (see Table A1 for details on all models and downloadable checkpoints).

[4]We use the pre-trained Glove embeddings trained on Common Crawl (840 billion tokens, 2.2 million vocab-size, cased, 300-dimensional vectors).

Table 2: Emb-GAM yields strong gains in validation accuracy for three datasets and is competitive for the two others. When restricted to only unigrams, the performance of Emb-GAM drops considerably.

|  |  | Financial phrasebank | Rotten tomatoes | SST2 | Emotion |
|---|---|---|---|---|---|
| Emb-GAM | **Emb-GAM** | **92.8%** $\pm$ 0.37% | **81.6%** $\pm$ 0.05% | **86.9%** $\pm$ 0.1% | **89.5%** $\pm$ 0.03% |
|  | Emb-GAM (unigrams only) | 86.4% $\pm$ 0.13% | 76.8% $\pm$ 0.19% | 81.7% $\pm$ 0.07% | 87.2% $\pm$ 0.06% |
| Interpretable baselines | Bag of ngrams | 85.0% $\pm$ 0.11% | 75.0% $\pm$ 0.09% | 82.8% $\pm$ 0.0% | 89.0% $\pm$ 0.09% |
|  | TF-IDF | 84.9% $\pm$ 0.16% | 75.9% $\pm$ 0.06% | 83.4% $\pm$ 0.11% | 89.2% $\pm$ 0.04% |
|  | GloVe | 80.5% $\pm$ 0.06% | 78.7% $\pm$ 0.03% | 80.1% $\pm$ 0.1% | 73.1% $\pm$ 0.09% |
| Black-box baseline | **BERT finetuned** | **98.0%** | **87.5%** | **92.4%** | **93.6%** |

**Comparing Emb-GAM performance with a black-box baseline**  In the studied data sets, the black-box baseline (a BERT finetuned model) outperforms Emb-GAM by 4%-6% accuracy. This is potentially a reasonable tradeoff in settings where interpretability, memory, or speed are critical. In cases involving inference memory/speed, Emb-GAM can be converted to a dictionary of coefficients roughly the size of the number of ngrams that appeared in training (see Table 1); for a trigram model, this yields roughly a 1,000-fold reduction in model size (compared to the ∼110 million trainable parameters in BERT), with much room for further size reduction (e.g. simply removing coefficients for trigrams that appear only once yields another 10-fold size reduction). Inference is nearly instantaneous, as it simply requires looking up coefficients in a dictionary and then a single sum (and does not require a GPU).

**Using Emb-GAM together with a black-box baseline**  In some situations, it may be useful to use Emb-GAM on some fraction of samples (for interpretability/memory/speed) but relegate the remaining samples to a black-box model. Here, we first predict each sample with Emb-GAM, then assess its confidence (how close its predicted probability for the top class is to 1). If it is above a pre-specified probability threshold, we use the Emb-GAM prediction. Otherwise, we compute the sample's prediction using a finetuned BERT model. Fig 3 shows the validation accuracy for the entire dataset as we vary the percentage of samples predicted with Emb-GAM. Since Emb-GAM yields probabilities which are reasonably calibrated (see Fig A2), rather than the accuracy linearly interpolating between Emb-GAM and BERT, a large percentage of samples can be predicted with Emb-GAM while incurring little to no drop in accuracy. For example, when using Emb-GAM on 50% of samples, the average drop in validation accuracy is only 0.0053.

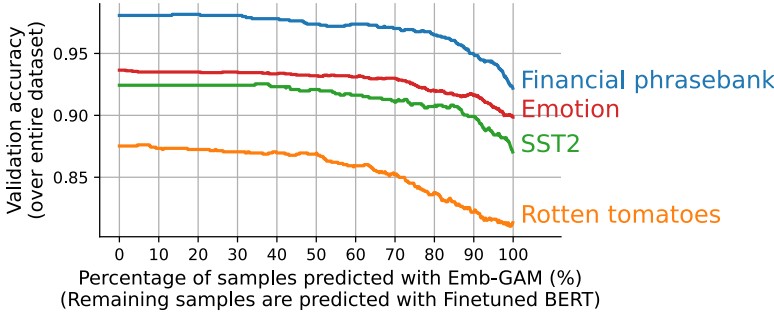

Figure 3: Accuracy when using Emb-GAM in combination with BERT. A large percentage of samples can be accurately predicted with Emb-GAM.

**Varying Emb-GAM model settings**  Table 3 shows how generalization accuracy changes when the LLM used to extract embeddings is varied between BERT, DistilBERT (Sanh et al., 2019) and RoBERTa (Liu et al., 2019). Across the variations, embeddings from finetuned models yield considerably better results than embeddings from non-finetuned models. We also investigate using different layers and ngram selection techniques in Table A2.

Table 3: Generalization accuracy varies depending on the model used to extract embeddings. Fine-tuning the embedding model improves Emb-GAM performance, and using DistilBERT lowers performance. Top two methods in each column are bolded.

|  | Financial phrasebank | Rotten tomatoes | SST2 | Emotion |
|---|---|---|---|---|
| BERT finetuned | **92.8%** $\pm$ 0.37% | **81.6%** $\pm$ 0.05% | 86.9% $\pm$ 0.1% | **89.5%** $\pm$ 0.03% |
| BERT | 84.1% $\pm$ 0.08% | 78.1% $\pm$ 0.16% | 82.8% $\pm$ 0.27% | 67.1% $\pm$ 0.06% |
| DistilBERT finetuned | 85.8% $\pm$ 0.34% | 78.5% $\pm$ 0.34% | 81.7% $\pm$ 0.07% | 68.8% $\pm$ 0.11% |
| DistilBERT | 81.7% $\pm$ 0.34% | 79.8% $\pm$ 0.08% | 86.8% $\pm$ 0.1% | 87.5% $\pm$ 0.11% |
| RoBERTa finetuned | 77.8% $\pm$ 0.31% | **83.6%** $\pm$ 0.03% | **89.1%** $\pm$ 0.24% | 88.5% $\pm$ 0.19% |

### 4.3 INTERPRETING A LEARNED MODEL

In this section, we investigate how to interpret a fitted Emb-GAM model, focusing on the *SST2* dataset, where the task is to classify whether a movie review's sentiment is positive or negative. We inspect an Emb-GAM model fitted using unigram and bigram features extracted from the BERT finetuned model; this model achieves 84% validation accuracy.

**Learned coefficients match human sentiment scores**  A trained Emb-GAM model can be interpreted for a single prediction (i.e. getting a score for each ngram in a single input, as in Fig 1) or for an entire dataset (i.e. by inspecting its learned coefficients). Fig 4A visualizes the learned Emb-GAM coefficients (i.e. the contribution to the prediction $w^T \phi(x_i)$) with the largest absolute values across the SST2 dataset. To show a diversity of ngrams, we show every fifth ngram rather than just the top ngrams. The learned coefficients are semantically reasonable and many contain strong interactions (e.g. *not very* is assigned to be very negative whereas *without resorting* is assigned to be very positive). Note that this form of model visualization easily allows a user to audit and edit the model with prior knowledge, e.g. by altering a coefficient.

Fig 4B shows how well the learned Emb-GAM coefficients match human-labeled sentiment phrase scores for unigrams/bigrams in SST (note: these continuous scores are separate from the binary sentence labels used for training in the SST2 dataset). Both are centered, so that 0 is neutral sentiment and positive/negative values correspond to positive/negative sentiment, respectively. There is a strong positive correlation between the coefficients and the human-labeled scores.

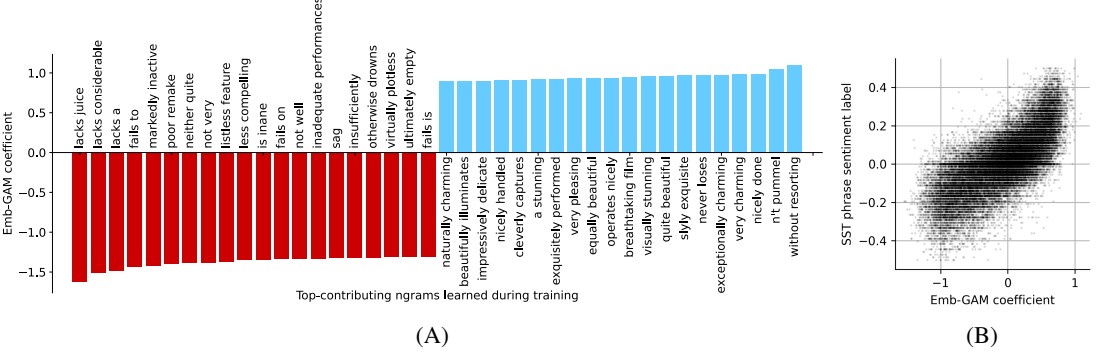

(A)          (B)

Figure 4: Top and bottom contributing ngrams to an Emb-GAM model trained on SST2 are (A) qualitatively semantically accurate and (B) match human-labeled phrase sentiment scores.

**Inferred coefficients for unseen ngrams match human sentiment scores**  One strength of Emb-GAM is its ability to infer linear coefficients for ngrams that were not seen during training. Whereas

baseline models generally assign each unknown ngram the same coefficient (e.g. 0), Emb-GAM can effectively assign these new ngrams accurate coefficients (as long as the new ngram consists of tokens in the vocabulary of the LLM being used).

As one example, Fig 5A shows that the Emb-GAM model trained only on bigrams in Fig 4 can automatically infer coefficients for *trigrams* (none of which were explicitly learned during training). The learned coefficients are semantically meaningful, even capturing three-way interactions, such as *not very amusing*. To show a diversity of ngrams, we show every 20th ngram rather than just the top ngrams. Fig 5B shows the learned trigram coefficients compared to the human-labeled SST phrase sentiment for all trigrams in SST. Again, there is a strong correlation, suggesting that the coefficients are semantically accurate.

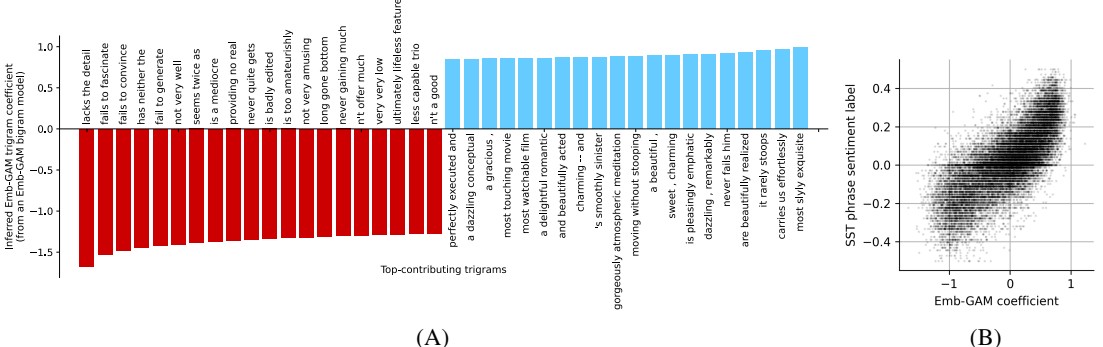

(A)                                                (B)

Figure 5: Inferred trigrams coefficients for a Emb-GAM model trained only on bigrams (A) qualitatively semantically accurate and (B) match human-labeled phrase sentiment scores.

**Test-time accuracy-interpretability tradeoff** The ability to effectively generalize to unseen tokens in Fig 5 raises the question of whether one can vary the order of ngrams used *at test-time*, to get a tradeoff between accuracy and interpretability (i.e. how many features are used to make a prediction). Fig 6A suggests this may be feasible; it shows the accuracy of an Emb-GAM model (using finetuned BERT) fitted using 4-grams as the order of ngrams used only for testing is varied. As the number of features used for testing increases, the performance tends to increase but interpretations become more difficult. Depending on the relative importance of accuracy and interpretability context, one may select to use a different number of features for testing.

Fig 6B characterizes the full tradeoff between the number of ngrams used for fitting and for testing for SST2. Generally, the best performance is achieved when the same number of ngrams is used for training and testing (the diagonal). Performance tends to degrade significantly when fewer ngrams are used for testing than training (lower-left). Results for all datasets show similar patterns (see Fig A1).

**Comparison with post-hoc feature importance** The coefficients learned by Emb-GAM often differ from importances assigned by post-hoc feature-importance methods. Emb-GAM learns a single coefficient for each ngram across the dataset, allowing for auditing/editing the model with visualizations such as Fig 4. In contrast, popular methods for post-hoc feature importance, such as LIME (Ribeiro et al., 2016) and SHAP (Lundberg & Lee, 2016) yield importance scores that vary based on the context in each input. This can be useful for debugging complex nonlinear models, but these scores (i) are approximations, (ii) must summarize nonlinear feature interactions, and (iii) vary across predictions, making transparent models preferable whenever possible.

Fig 7 shows an example of the Emb-GAM coefficients for the SST2 model from Fig 4 for different ngrams when making a prediction for the phrase *not very good*. While Emb-GAM yields scores for each subphrase that match human judgement (as seen in Fig 4B and Fig 5B), posthoc feature importance methods summarize the interactions between different ngrams into individual words, potentially making interpretation difficult. Scores are rescaled to be between -1 and 1 to make them comparable. See Emb-GAM scores for many top-interacting phrases in Fig A3.

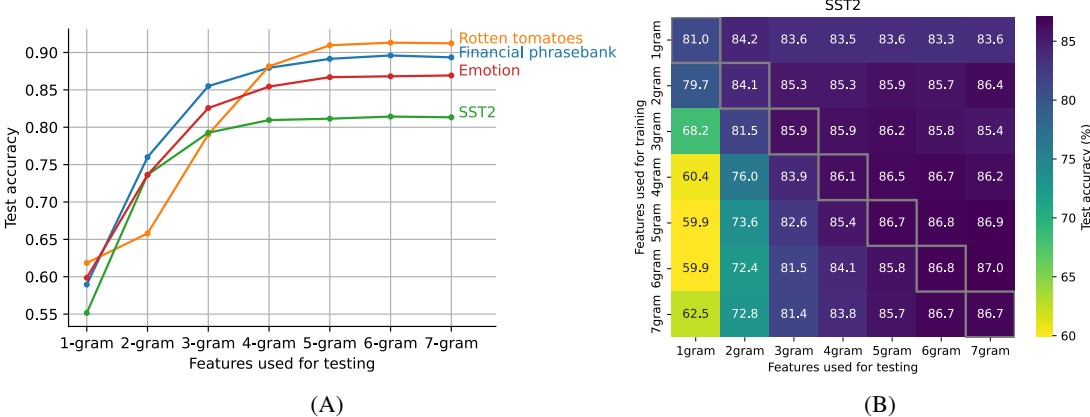

Figure 6: Emb-GAM performance when varying the number of ngrams used for *testing*. (A) Performance for a model fitted using 4-grams. (B) Full breakdown when varying the ngrams used for training and testing.

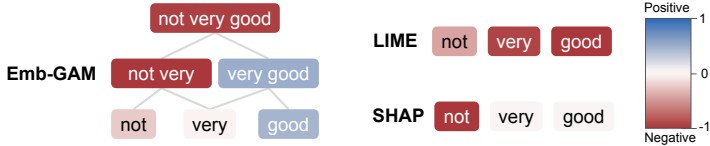

Figure 7: Comparing Emb-GAM ngram coefficients (left) to word-level feature importances from posthoc methods (right): LIME and SHAP.

## 5    DISCUSSION

Emb-GAM helps bridge the gap between black-box models and transparent models in NLP in terms of accuracy and speed. This potentially opens the door for introducing NLP in high-stakes domains, such as medical decision-making and on compute-limited hardware. While the work here focuses on NLP, the same pipeline could be applied to other domains wherever effective language models are available to extract meaningful embeddings (e.g. computer vision).

One limitation of this work is that performance does not improve unless using an LLM finetuned on a downstream dataset. Future work could train the entire Emb-GAM pipeline end-to-end, potentially improving the embedding representation. Such a representation could allow Emb-GAM to outperform even a black-box finetuned neural network, as the GAM inductive bias may help prevent overfitting in data-limited settings.

There are many potentially useful extensions of Emb-GAM. One notable extension would build on the nonlinearity present in GAMs such as the explainable boosting machine (Caruana et al., 2015), to nonlinearly transform the embedding for each ngram with a model before summing to obtain the final prediction.

Finally, there are many ways Emb-GAM can be improved and used in the real-world. When domain knowledge is available, more meaningful inputs can be used to extract ngrams or to make a sparser model. Additionally, compression techniques can be applied to the Emb-GAM model after converting to a dictionary of coefficients to reduce memory requirements. Finally, Emb-GAM can also be used for tasks beyond classification, such as sequence prediction or outlier detection. We hope that the introduction of Emb-GAM can help push improved performance prediction into high-stakes applications in the real world and reduce unnecessary energy/compute usage.

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
