# OpenReview forum: "Emb-GAM: an Interpretable and Efficient Predictor using Pre-trained Language Models"
_ICLR.cc/2023/Conference — Submitted to ICLR 2023_

### Official Review · Reviewer_Ks5M · 2022-10-23

**Confidence:** 5
**Correctness:** 3
**Technical Novelty And Significance:** 2
**Empirical Novelty And Significance:** 1
**Recommendation:** 3

**Clarity, Quality, Novelty And Reproducibility:**

Overall, the quality of the paper is below the bar of ICLR, the novelty is not enough.

**Strength And Weaknesses:**

Strengths
- The paper is easy to read.

Weaknesses
- Only compare with two basic baselines.
- Only evaluate on classification datasets.
- The idea is not novel. The procedure of Emb-GAM is already common especially in cross-domain retrieval.
- The length of ngram still need to be specified. And this matters. As higher-order ngrams are added, the computation also explodes.
- It depends on language model. For each new dataset, finetuning is needed.
- I have doubt about using a linear model to learn high-dimensional feature interaction.

**Summary Of The Paper:**

This paper proposes a simple and interpretable additive model called Emb-GAM by leveraging pretrained language model.

**Summary Of The Review:**

Although this paper handles a very important task, but the idea and evaluation are both not enough. Besides, there are many limitations as mentioned in the paper. Therefore, I tend to reject this paper.

---

> ### Author Response · Authors · 2022-11-15
> **Thanks for your comments**
>
> Thank you for your thoughtful comments, we sincerely appreciate the time you put into them and have significantly improved the results and manuscript in response. Point-by-point responses below:
>
> > Only compare with two basic baselines.
>
> We've added 2 new comparisons.
>
> First, we added a comparison to GloVE embeddings to the results table + a discussion of related non-contextualized word embeddings in the related work section. The use of an LLM for extracting ngram embeddings provides nontrivial advantages over explicitly training word/ngram embeddings, e.g. (1) it doesn't require retraining a model and  (2) effectively scales to relatively long ngrams (length 7 in our experiments, much larger than people do for ngram models).
>
> Second, we've added comparisons to the finetuned BERT model in Tab 2 and Fig 3. We find that finetuned BERT outperforms Emb-GAM by 5-6% on average, potentially a reasonable tradeoff in settings where interpretability, memory, or speed are critical. Moreover, since the Emb-GAM model is reasonably calibrated, it can be used on some fraction of samples (for interpretability/memory/speed) but relegate the remaining samples to a black-box model. For example, when using Emb-GAM on 50% of samples, the average drop in validation accuracy is only 0.5%.
>
> > The idea is not novel. The procedure of Emb-GAM is already common especially in cross-domain retrieval
>
> Could you point us to a reference for this? We are unable to find such works.
>
> > The length of ngram still need to be specified. And this matters.
>
> Indeed the ngram length must be specified (we choose it here via cross-validation), but we have added some results in Fig 6 which show that the performance is not too sensitive to the ngram length. Even more, the ngram length can be varied at test time with little impact on performance.
>
> > As higher-order ngrams are added, the computation also explodes.
>
> While computation grows with the order of the ngrams, unlike models like bag-of-words/GloVe/ELMo, computation does not explode with the ngram order (in the worst case it grows linearly with the order).
>
> > It depends on language model. For each new dataset, finetuning is needed.
>
> Indeed this is currently a weakness, but also potentially a strength, as it allows for integrating information about a label into the representation in a way that unsupervised methods cannot.
>
> > I have doubt about using a linear model to learn high-dimensional feature interaction.
>
> While Emb-GAM can't solve all NLP problems, there are a large class of problems for which speed or interpretability are important and Emb-GAM can help in these situations.
>
> > the novelty is not enough.
>
> Given that the experimental gains are meaningful, we feel that the simplicity of our method should be an argument for its acceptance, not against. In machine learning, there is often the temptation to propose complicated “highly novel” approaches, many of which are never adopted. In contrast, we explicitly tried to identify the simplest approach that could produce a meaningful improvement over the current SOTA. In interpretability, we feel simplicity is particularly important, as we must consider the human component in providing simple, easy to understand insights into the model.

---

### Official Review · Reviewer_HyrP · 2022-10-24

**Confidence:** 3
**Correctness:** 2
**Technical Novelty And Significance:** 1
**Empirical Novelty And Significance:** Not applicable
**Recommendation:** 1

**Clarity, Quality, Novelty And Reproducibility:**

# Clarity, Quality, Novelty, and Reproducibility
## Clarity
The paper is, in general, well written and clear.

## Quality
This paper doesn't have any points that are (to my knowledge) technical incorrect, but it does seem to be leveraging large language models for a not well-motivated use case (namely, to produce minimally  or non-contextualized static embeddings).

## Novelty
This paper does not have any significant sources of novelty. While I'm not aware of any papers that use GAMs with static BERT embeddings explicitly, using simpler models with static embeddings is not a novel concept; indeed, any use of word2vec with an n-gram pooling operation would qualify as such.

## Reproducibility
I think this paper would be relatively reproducible, especially as the authors commit to releasing code on github.


**Strength And Weaknesses:**

# Strengths and Weaknesses

## Key Strengths (reasons I would advocate this paper be accepted)
None

## Key Weaknesses (reasons I would advocate this paper be rejected; unordered)
  1. This approach is insufficiently technically sophisticated or novel. Ultimately, all that you are doing is producing n-gram embeddings using a LLM, then training a traditional, low-capacity model on those embeddings. This is not a new idea (especially as you produce non-contextualized word embeddings, this is equivalent in terms of technical sophistication to prior, static embedding approaches such as word2vec, elmo, etc.), and your experiments likewise do not produce dramatically novel findings to fit the ICLR venue.

  2. This approach does not seem well founded. You propose feeding _isolated_ n-grams through a large language model (like BERT) to produce n-gram embeddings. But, large language models like BERT are worthwhile over static n-gram embedding approaches precisely because they integrate longer-range context into their embeddings (thereby producing contextualized word embeddings). Under your system, you're explicitly producing _non-contextual_ embeddings using a _contextualized_ approach, which is slightly problematic. It certainly isn't likely to not work, but the significant expense of a system like BERT seems wasted in your setting and warrants more justification than you give it, both qualitatively and quantitatively. Furthermore, this also actually subverts your claim of interpretability -- While I understand that BERT embeddings are frozen prior to your final use via the GAM model, it still isn't the case that you could easily answer the question of why one n-gram is preferred over another given your interpretability analysis. With a bag-of-ngrams representation and linear coefficient based interpretability study, you can reason that one n-gram is preferred over another because it is more correlated with the output in question in your dataset; however, by using BERT to extract features, if I see from the GAM approach that "very good" and "super great" have wildly different final coefficients, I can't reason about why this is. Thus, it isn't clear how your approach is likely to solve the problem you identify; namely, producing interpretable, efficient predictions while leveraging the advantages of a DNN.

  3. You do not sufficiently justify your model empirically to motivate its use. In particular,
   - You lack sufficient comparisons against baselines, as you only compare against bag of n-grams and TF-IDF. You don't look at nearest neighbor approaches, methods leveraging simpler static embeddings like  word2vec, or existing, post-hoc interpretability methods.
   - You don't quantify the performance gap your interpretable method induces relative to high-capacity system. if BERT (as traditionally used) performs dramatically better on these tasks compared to your method, that is relevant to the overall impact of this work.
   - You don't attempt to quantify the gain in interpretability your approach offers over existing interpretability methods. As interpretability is the primary goal of your work, this omission is a major lack.

## Minor Strengths (things I like, but wouldn't sway me on their own)
  1. I think interpretability for validation/debugging/debiasing purposes is very useful, and so work in that space is valuable. This is only a minor strength as I'm not sure your approach can meaningfully contribute towards the validation/debugging/debiasing goals given it proposes a fully new model.
  2. The paper is, in general, well written.

## Minor Weaknesses (things I dislike, but wouldn't sway me on their own)
None


**Summary Of The Paper:**

# Summary
Emb-GAM: an Interpretable and Efficient Predictor using Pre-trained Language Models

## What is the problem?
Deep neural networks for NLP are high-performing, but uninterpretable. They are also computationally more intensive.

## Why is it impactful?
Some people argue that intepretability is essential to produce trust in models in high-stakes domains. However, it is important to note that this view has some detractors, https://hai.stanford.edu/news/should-ai-models-be-explainable-depends. A possibly more universal argument in favor of interpretability is that interpretability is valuable as a model validation, debugging, and de-biasing tool.

I highlight these two different views as your method would, I believe, only help support one of them (the former), not the latter.

## Why have existing approaches failed?
The authors only comment briefly on prior approaches for producing interpretable DNN-like models, but argue that they lose some information by summarizing the model, or fail to explain the model as a whole.

## What is this paper's contribution?
The authors propose to use a DNN to produce feature embeddings of n-grams in text, then to use a GAM to provide a low-capacity, efficient, interpretable output prediction of the final task given these LLM contextualized embeddings.

## How do they validate their contributions?
The authors demonstrate the superiority of their approach against a Bag-of-ngrams and TF-IDF approach on 4 NLP datasets, and examine the qualitative semantic alignment between identified high-importance n-grams and the underlying task.


**Summary Of The Review:**

 Summary of the Review
  1. {Correctness}
     (incorrected or not at all supported - well-supported and correct)
     I'm scoring this a 2 here, as the results seem correct but the conclusions (that this model is an acceptable trade-off of interpretability, efficiency, and performance) don't seem well supported as interpretability is not quantitatively evaluated against any baselines and performance is not contrasted against underlying LLMs).
  2. {Technical Novelty and Significance}
     (neither significant nor novel - significant and do not exist in prior works)
     I'm scoring this a 1, as I don't feel this is novel nor does it show sufficient technical significance.
  3. {Empirical Novelty and Significance}
     (neither significant nor novel - significant and do not exist in prior works)
     Not Applicable
  4. {Flag for Ethics Review}
     No flag
  5. Recommendation
     (strong reject - strong accept)
     I'm recommending strong reject here as I don't think the approach is well justified, either conceptually or via their provided experiments, and it also doesn't meet the technical sophistication/novelty bar for ICLR.

## What would make me raise my score? (Things that you can do that would, pending their results and the manner in which you accomplish them, make me raise my score)
I would need to see (a) additional experiments establishing that this method does significantly better than comparable, simpler approaches, such as those leveraging word2vec or glove embeddings as the source of contextual embeddings in raw performance, (b) that it performs comparably or better than (on some scale, preferablly some kind of human evaluation) at interpretability than existing interpretability methods, including post-hoc methods like LIME and attentional analysis from BERT, and (c) that it doesn't perform too much worse than just using BERT alone that it is still a viable solution.

I suspect these experiments are too large in scope to fit within a revision, and would necessitate a full re-write. Additionally, even if those experiments were performed to my satisfication, I'm still not sure ICLR would be the right fit for this paper, and would probably recommend the authors look at a more focused NLP venue instead given the limited technical sophistication of the underlying methods here.

---

> ### Author Response · Authors · 2022-11-12
> **Thanks for your comments**
>
> Thanks for your thoughtful comments, we sincerely appreciate the time you put into them and have seriously reworked the results and manuscript in response. Point-by-point responses below:
>
> > “Some people argue that intepretability is essential to produce trust in models in high-stakes domains....A possibly more universal argument in favor of interpretability is that interpretability is valuable as a model validation, debugging, and de-biasing tool.
> >
> > I highlight these two different views as your method would, I believe, only help support one of them (the former), not the latter.”
>
> We agree that interpretability should be grounded in real-world improvements, such as validation/debugging. While producing trust is useful for enabling new usecases / regulatory concerns, the real strenghts of Emb-GAM are in enabling validation/debugging through inspection such as fig 4, 5, & 7. Ngrams can be easily evaluated to identify errors and linear coefficients can be edited to fix model issues.
>
> As a second motivation, Emb-GAM also provides dramatic reduction in size/latency for inference. We have reworked the manuscript to express this point clearly throughout.
>
> > This approach is insufficiently technically sophisticated or novel
>
> Given that the experimental gains are meaningful, we feel that the simplicity of our method should be an argument for its acceptance, not against. In machine learning, there is often the temptation to propose complicated “highly novel” approaches, many of which are never adopted. In contrast, we explicitly tried to identify the simplest approach that could produce a meaningful improvement over the current SOTA. In interpretability, we feel simplicity is particularly important, as we must consider the human component in providing simple, easy to understand insights into the model.
>
> > This approach does not seem well founded...Under your system, you're explicitly producing *non-contextual* embeddings using a *contextualized* approach
>
> We've added a comparison to GloVE embeddings to the results table + a discussion of related non-contextualized word embeddings in the related work section. The use of an LLM for extracting ngram embeddings provides nontrivial advantages over explicitly training word/ngram embeddings, e.g. (1) it doesn't require retraining a model, (2) effectively scales to relatively long ngrams (length 7 in our experiments, much larger than people do for ngram models) (3) it allows for integrating information about a label into the representation which we find to be crucial for performance, e.g. by finetuning the model (see Table 3). While there may be an alternative way to yield such a representation by training a new model/architecture, we find that the LLM approach is much simpler to carry out in practice.
>
> > it still isn't the case that you could easily answer the question of why one n-gram is preferred over another given your interpretability analysis
>
> This is true, this is potentially a limitation of our study if one was doing some kind of causal analysis (e.g. interpret the data) but does not limit our ability to interpret the model, as it can be decomposed exactly into an additive model.
>
> > You lack sufficient comparisons against baselines
>
> As noted, we've added a comparison to GloVe + a new fig 7 with a discussion comparing to post-hoc interpretability methods. While post-hoc interpretability methods are useful for debugging a complex nonlinear model, Emb-GAM coefficients are much simpler to interpret as they are constant across all predictions of the dataset.
>
> > You don't quantify the performance gap your interpretable method induces relative to high-capacity system
>
> We've added these comparisons in Tab 2 and Fig 3. We find that finetuned BERT outperforms Emb-GAM by 5-6% on average, potentially a reasonable tradeoff in settings where interpretability, memory, or speed are critical.
>
> Moreover, since the Emb-GAM model is reasonably calibrated, it can be used on some fraction of samples (for interpretability/memory/speed) but relegate the remaining samples to a black-box model. For example, when using Emb-GAM on 50% of samples, the average drop in validation accuracy is only 0.5%.

---

### Official Review · Reviewer_2xXf · 2022-10-24

**Confidence:** 4
**Correctness:** 3
**Technical Novelty And Significance:** 4
**Empirical Novelty And Significance:** 4
**Recommendation:** 5

**Clarity, Quality, Novelty And Reproducibility:**

The work is written clearly, novel and could be reproduced for the most part based on the writing of the paper ( hyper parameters for the GAM, what variant of BERT was used ( large, base, cased? ))

**Strength And Weaknesses:**

**Strengths:**
The paper is well written and situates itself well against the existing literature,
The idea is novel and there is need for interpretability in NLP.

**Weaknesses:**
However the performance of their model compared with deep learning models ( for which they aim to bridge the gap ) is not given ( ie, Emb-GAM is compared against simple, interpretable baselines, which is fine if one of the main claims of the paper is to bridge the gap in performance between GAMs and DNNs ).  How big is the gap?  A quick look shows for SST2,  it looks to be 8 to 10 points.  These results need to be included and discussed in the paper.

Also the ablation with respect to using DistilBERT or fine-tuning and the pooling strategy could have been relegated to an appendix in lieu of other experiments (or one of the Figures from Appendix Figure 1a).  For instance, DistilBERT is an optimized/pruned version of BERT whose advantage is that its smaller, parameter wise while getting near BERT performance, so using it will definitely not improve results and the speed gain is negligible in your model ( which uses BERT solely as an embedder ).  It would have been better to use DeBERTa or RoBERTa for comparisons.

The utility of the interpretability of the coefficients in the experiments could have been made stronger by a human study of the learned coefficient as opposed to showing global coefficient values for bi-grams / trim-grams and saying they seem qualitatively reasonable.

Also the argument for the learning of interactions via ngram embeddings was a little unconvincing/unclear in that I’m not sure the use of 10 bigrams and their constituent unigrams was the best argument here.  Yes the Emb-GAM model’s bigram coefficient is not the simple sum of unigram coefficients its learned, but its not clear why/how that suggests the model has successfully learned interactions?
The sum of the embedding for word1  and the embedding for word2 is not equal to the embedding of the combined word1 and word2  so it stands to reason a GAM will have different coefficients for all 3.   Why not compare against LIME or SHAP at the individual level since this model purports to handle interactions better and show it to humans for quality comparisons?  A better case needs to be made for how/when to use this method if the performance gap with deep learning models is high and post hoc feature attribution methods are available ( although I agree with all the possible limitations listed of using post-hoc methods )



**Summary Of The Paper:**

The authors identify that deep learning models (DLM)  give state of the art performance on NLP tasks, but are black box models which are not interpretable in themselves.  They propose using a generalized additive model (GAM) setup which uses the sum of ngram embeddings for text classification tasks to bridge the gap between DLM performance and GAM interpretability.  Their Emb-GAM model is interpretable and decomposable because a prediction is a linear combination of n-gram embeddings fed into a softmax or logit function.  They show the model’s accuracy performance compared with two interpretable baselines ( bag of words and TF-IDF ) on 5 datasets, along with ablations on what variant of BERT and BERT output to use and the effect of the number of n-gram features used by the model ( generally the higher the n, the higher task accuracy and the potentially less interpretable ).  They also show how to interpret model results by inspecting n-gram coefficients at a global level and how the model can infer the importance of unseen n-grams during training at test time.

**Summary Of The Review:**

The idea is novel and there is need for interpretability in NLP, but the performance of their model compared with deep learning models which the authors claim to bridge is missing and needs to be a part of this work and discussion.
Some ablations, experiments and analysis of the model could have been improved/added to strengthen the argument for what is an interesting and potentially compelling line of research.

---

> ### Author Response · Authors · 2022-11-15
> **Thank you for your comments**
>
> Thank you for your thoughtful comments, we sincerely appreciate the time you put into them and have significantly improved the results and manuscript in response. Point-by-point responses below:
>
> > the performance of their model compared with deep learning models ( for which they aim to bridge the gap ) is not given
>
> We've added 2 new comparisons.
>
> First, we added a comparison to GloVE embeddings to the results table + a discussion of related non-contextualized word embeddings in the related work section. The use of an LLM for extracting ngram embeddings provides nontrivial advantages over explicitly training word/ngram embeddings, e.g. (1) it doesn't require retraining a model and  (2) effectively scales to relatively long ngrams (length 7 in our experiments, much larger than people do for ngram models).
>
> Second, we've added comparisons to the finetuned BERT model in Tab 2 and Fig 3. We find that finetuned BERT outperforms Emb-GAM by 5-6% on average, potentially a reasonable tradeoff in settings where interpretability, memory, or speed are critical. Moreover, since the Emb-GAM model is reasonably calibrated, it can be used on some fraction of samples (for interpretability/memory/speed) but relegate the remaining samples to a black-box model. For example, when using Emb-GAM on 50% of samples, the average drop in validation accuracy is only 0.5%.
>
> >  the ablation with respect to using DistilBERT or fine-tuning and the pooling strategy could have been relegated to an appendix in lieu of other experiments
>
> Thanks for this recommendation - we have moved the layer/strategy ablations to the appendix and added a comparison with RoBERTa to Table 3.
>
> > The utility of the interpretability of the coefficients in the experiments could have been made stronger by a human study of the learned coefficient as opposed to showing global coefficient values for bi-grams / trim-grams and saying they seem qualitatively reasonable.
>
> Thank you for this suggestion. We've added two new panels (Fig 4B and Fig 5B) showing the relationship between the Emb-GAM coefficients and the sentiment phrase-label scores from SST (which show strong agreement).
>
> > Also the argument for the learning of interactions via ngram embeddings was a little unconvincing/unclear in that I’m not sure the use of 10 bigrams and their constituent unigrams was the best argument here.
>
> Thanks, we have moved the bigram barplot to the appendix and replaced it instead with fig 7 and added text discussing the relationship between the learned coefficients and how they related to pos-hoc interpretation scores such as LIME/SHAP (the biggest difference being that they are non-contextualized, making them constant across all predictions in the dataset).
>
> > A better case needs to be made for how/when to use this method if the performance gap with deep learning models is high
>
> We've rewritten the text throughout to emphasize when Emb-GAM is useful, particularly emphasizing situations with requirements for low-memory/fast inference in addition to interpretability.

---

### Decision · Program_Chairs · 2023-01-20

**Decision:**

Reject

**Justification For Why Not Higher Score:**

This is the beginning of a good paper, but the changes required were significant and though the authors made a valiant effort to meet these requests we believe a new round of review is required.

**Justification For Why Not Lower Score:**

n/a

**Metareview: Summary, Strengths And Weaknesses:**

This paper contributes to the ongoing conversation around model interpretability in NLP.  The reviewers all agreed that the topic is important, and that there were some interesting contributions in this paper.   The AC appreciates the authors' commitment to the review process, their responses to the reviews, and their additions to the paper during the review period.  I think reviewer HyrP's comments best capture the general feelings of the reviewers.  The original paper lacks sufficient baselines, and though the revised paper contains some new baselines, the changes are significant enough that 1) the paper may need to be reworked to be more compelling and 2) the changes may be enough to require a re-review.  There is solid work here, and I agree with reviewers who encourage the authors to consider other venues where this work may be more appreciated.